# Semi-Supervised Portrait Matting via the Collaboration of Teacher–Student Network and Adaptive Strategies

**Xinyue Zhang** , **Guodong Wang \***, **Chenglizhao Chen, Hao Dong** and **Mingju Shao**

College of Computer Science and Technology, Qingdao University, Qingdao 266071, China
\* Correspondence: doctorwgd@gmail.com

**Abstract:** In the portrait matting domain, existing methods rely entirely on annotated images for learning. However, delicate manual annotations are time-consuming and there are few detailed datasets available. To reduce complete dependency on labeled datasets, we design a semi-supervised network (ASSN) with two kinds of innovative adaptive strategies for portrait matting. Three pivotal sub-modules are embedded in our architecture, including a static teacher network (S-TN), a static student network (S-SN), and an adaptive student network (A-SN). S-TN and S-SN are modules that need to be trained with a small number of high-quality labeled datasets. Moreover, A-SN and S-SN share the same module parameters. When processing unlabeled datasets, A-SN adopts the adaptive strategies designed by us to discard the dependence on labeled datasets. The adaptive strategies include: (i) An auxiliary adaption: The teacher network with complicated design not only provides alpha mattes for the adaptive student network but also transmits rough segmentation results and edge graphs as optimization reference standards. (ii) A self-adjusting adaption: The adaptive network can make self-supervised to the characteristics of different layers. In addition, we have produced a finely annotated dataset for scholars in the field. Compared with existing datasets, our dataset complements the following two types of data neglected in previous datasets: (i) Images taken by multiple people. (ii) Images under low light conditions.

**Keywords:** portrait matting; semi-supervised; attention mechanism; adjustment strategy

## 1. Introduction

Portrait matting is an extraordinary image processing task in the computer version. Its core goal is to predict accurate alpha mattes that can be used to capture the foregrounds of images. Portrait matting is mainly used for background replacement. Since it is difficult to produce hair-fine mattes, the task has always faced the problem of how to improve the accuracy of the effect in the case of insufficient datasets.

Existing methods [1,2] have also made progress with only small amounts of labeled datasets. However, in the face of unlabeled datasets, the effect of the existing models cannot be improved, because existing portrait matting models rely entirely on labeled data for training and cannot adjust on unlabeled datasets. In the existing portrait matting methods, the trained model can no longer improve its performance when faced with unlabeled datasets. This limitation greatly reduces the generalization ability of the model over unlabeled datasets. Recently, semi-supervised networks have attracted the attention of many tasks, such as knowledge distillation. Knowledge distillation is the transfer of knowledge by introducing the teacher network the ability to acquire the soft targets and then inducing a student network to conduct training. It has been widely utilized in many computer vision tasks and achieved considerable results; however, in the case of portrait matting, a task that requires hair-precision results, the knowledge distillation should not focus on allowing the student network to simply learn the soft label that the teacher network outputs at the end. Some characteristics of the intermediate-level output of the teacher network should also enjoy the treatment of being transmitted to the student network for

learning. In addition, to avoid excessive dependence on the features provided by the teacher network in the training process, the student network should also have the ability of self-supervision and adjustment.

Therefore, to achieve a performance-friendly portrait matting algorithm in the case of insufficient datasets, we constructed a semi-supervised network (ASSN) based on the idea of knowledge distillation. In addition, we designed two adaptive strategies to assist semi-supervised networks in dealing with the unlabeled datasets. On the one hand, the student network constructed by us is supervised by segmentation results, edge graphs, and alpha mattes generated in the teacher network for learning; on the other hand, the student network also directly supervises different layers of the self-network to avoid excessive dependence on the teacher network. Compared with existing fully supervised learning networks in the portrait matting field, the semi-supervised network with adaptive strategies can obtain further improvement space and stronger generalization ability on unlabeled datasets. Our main contributions are as follows:

- A concrete instantiation of the semi-supervised network architecture (called ASSN). The architecture consists of three sub-networks, namely, a static teacher network (S-TN), a static student network (S-SN), and an adaptive student network (A-SN). Among them, the adaptive student network is the final applied lightweight network, which successfully acquires the ability to further improve performance on unlabeled data with the assistance of the static teacher network.
- Two adaptive strategies to improve generalization on unlabeled datasets (as shown in Figure 1). Firstly, the auxiliary adaption ensures that the student network is not only supervised by the alpha mattes generated by the teacher network but also needs to receive the characteristics obtained in the middle layer of the network, including the segmentation results and edge graphs. Second, the self-adjusting adaption guarantees the similarity comparison between the characteristics of different levels in the student network.
- Twenty-four groups of comparative experiments and several groups of ablation experiments are performed on several datasets.
- An elaborate hand-annotated dataset has been produced and will be available to scholars in this field. We supplemented two types of images that are missing from existing datasets: images from multiple people and images taken in low-light conditions.

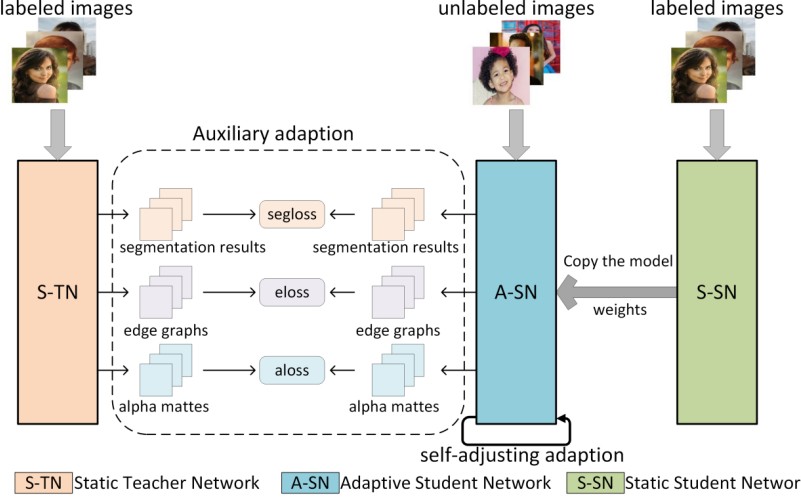

**Figure 1.** A schematic illustration of the collaborative process between our designed semi-supervisory network and two adaptive strategies.

## 2. Related Work

Three related areas of research are described in the following, including portrait matting, knowledge distillation, and semi-supervised.

## 2.1. Portrait Matting

Although some image segmentation methods can also recognize human contours, these contours are usually output as binary masks. Unlike image segmentation, portrait matting requires more accurate alpha mattes. Portrait matting can be divided into three categories [3] based on the input needed to enter into the network: (i) Trimap-based matting. These methods [4,5] require an input of trimaps in addition to the given images containing the portraits. Trimaps are manually annotated in three different colors. It is a rough partition of a given image, including, the foreground, background, and unknown region to be solved. (ii) Background-based matting. These methods [6–9] predict the portrait areas with the help of background images and initial images with people. Although it is more advantageous to identify portrait areas with given background images, background matting is not suitable for those with dynamic backgrounds. (iii) Auxiliary-free matting methods [10–16] do not require additional input except for images to be identified. These approaches, which require no additional input other than images, are attracting more and more scholars' attention. It does not require complex trimaps, nor does it require background images. A trained network can recognize portrait areas by simply sending the portrait images.

## 2.2. Knowledge Distillation

Knowledge distillation is a model compression method, which is a training method based on the idea of a teacher–student network [17–20]. Because of its simplicity and effectiveness, it is widely used in the industry. The process of knowledge distillation [21–25] is divided into two stages: (i) A "teacher model" (NET-T). This is characterized by a relatively complex model and can also be integrated by multiple separately trained models. For the "teacher model", we do not need to make any restrictions on model architecture, the number of parameters, and integration. The only requirement is that for input X, it can output Y, where Y is mapped by SoftMax, and the output value corresponds to the probability value of the corresponding category. (ii) A "student model" (NET-S). This is a single model with a small number of parameters and a relatively simple model structure. Similarly, for input X, it can output Y, and Y can also output the probability value corresponding to the corresponding category after SoftMax mapping. Since we already have a Net-T with strong generalization ability, we can directly let Net-S learn the generalization ability of Net-T. In this paper, our approach differs from existing ideas of knowledge distillation in the following ways. First, knowledge distillation is only an idea of using a large model to guide a small model to learn, rather than a concrete framework. Based on this idea, we designed a specific architecture suitable for the field of portrait matting. Under the same training conditions, it is better than the existing methods of portrait matting. Secondly, the object adopted for guidance is different from existing knowledge distillation methods. In our auxiliary adaption and self-adjusting adaption, we specify the segmentation results, edge graphs, and alpha mattes of the network for guidance. It is not guided by some feature layer or final output as existing methods are. Finally, as shown in Figure 1, our two adaptive strategies are combined with the semi-supervised network. This design enables our architecture not only to be trained on labeled datasets but also to be further improved when faced with unlabeled datasets.

## 2.3. Semi-Supervised

Semi-supervised learning is used to make the learner use unlabeled datasets to improve learning performance [26]. In semi-supervised learning, there are both labeled datasets and unlabeled datasets. Generally, the quantity of unlabeled datasets is much larger than labeled datasets. In general, a good predictive model is obtained by making full use of labeled and unlabeled datasets. This operation [27–29] enables the network not only to obtain the optimal prediction of unknown datasets but also to obtain high generalization ability. Since the alpha mattes required for training need meticulous manual annotation

and the available amount is small, it is necessary to realize portrait matting employing semi-supervised learning.

## 3. Proposed Method

To overcome the performance limitation of portrait matting after training with a small number of label datasets, we design a semi-supervised network based on the idea of knowledge distillation and propose two kinds of adaptive strategies when the network faces unlabeled data. It is worth noting that compared with previous knowledge distillation methods, our network has the following three innovations: First, our model can further improve performance when faced with unlabeled datasets. Since the teacher model is more complex than the student model, the prediction results of the teacher model are relatively more precise and can be utilized to guide the student model in training. The segmentation results, edge graphs, and alpha mattes generated by the teacher model are adopted as pseudo-labels in the training of the student model. With the help of these pseudo-labels, the student model can further improve its performance on unlabeled datasets. Secondly, our student network not only fully relies on the characteristics generated by the teacher network for learning, but also carries out the supervision between different levels in the self-network. Finally, our teacher network adopts the design of two backbones, and the knowledge learned by one backbone is constantly used as a residual margin to supplement the extracted features of the other backbone.

In this section, we describe the core proposed instantiations of the overall architecture and the two kinds of adaptive strategies.

### 3.1. Baseline Network Architecture

Our network structure can be roughly divided into three sub-networks: (1) A static teacher network (S-TN) with a complex network structure design. (2) A simple static student network (S-SN) pruned based on the teacher network. (3) An adaptive student network (A-SN) that copies the parameters of the static student network model. Both the static teacher network and the static student network will carry out a certain number of pre-training steps on the labeled datasets. After the training, the saved training parameters will not change, so we call these two networks static. The motivations of the three sub-networks are as follows: the static teacher network is designed to guide the student network for training; the static student network is employed for training on labeled data; the adaptive student network is used to achieve improved results on unlabeled data by combining the two types of adaptive strategies. After the training, we copy the parameters of the static student network to an adaptive student network. When faced with some unlabeled data, we adjust network parameters using two strategies of auxiliary adaption and self-adaptive adaption to achieve semi-supervised learning. In the following paragraphs, we introduce the architecture of the static teacher network and how to prune it to obtain the static student network structure. We first introduce the individual modules of the network in the first three parts, then introduce the process of data transmission in the static teacher network in the fourth part, and finally introduce how to obtain the static student network by pruning the static teacher network.

#### 3.1.1. Backbone

In our training network, we utilized the layout of two backbones. FBNetV2 [30] was selected as our backbone1, and MobileNetv3 [31] was embedded as backbone2. It has been proved in many existing papers [32–35] that the effect of a network can be improved by supplementing the original primary features with residual edges after continuously extracting abstract features. Different backbone focuses on extracting different features, so we continue to obtain high-level abstract features from backbone2 and use the primary features extracted from backbone1 to supplement. As can be seen from Figure 2, the features obtained from backbone2 have undergone more complex processing, so these features are more abstract than those obtained directly from backbone1. Based on experimental data

(as shown in Table 1), we deployed FBNetV2 [30] as backbone1 and MobileNetV3 [31] as backnone2. In the MobilenetV3 network, the substructure modules are arranged in parallel, while FBNetV2 is arranged in series. In terms of the diversity and richness of the features, the results of the intermediate layer of the network are far inferior to those of the last layer of the network because of the series structure of the network. In the parallel architecture, each branch can acquire unique characteristics due to the different embedded modules. Design differences between branches result in certain differences in the characteristics of each branch. The complementarity between branches and further feature extraction promote the establishment of richness. In the series structure, however, the character is simply enriched as the network deepens. The characteristics of diversity and richness in the parallel structure are more suitable for the embedded network for deep information mining. Therefore, compared with FBNetV2, mobilnetV3 is more suitable to output multiple results from the parallel structure to guide the student network to learn from unlabeled data. So, MobilenetV3 was embedded as the backbone2 in the network, rather than FBNetV2 as the backbone2.

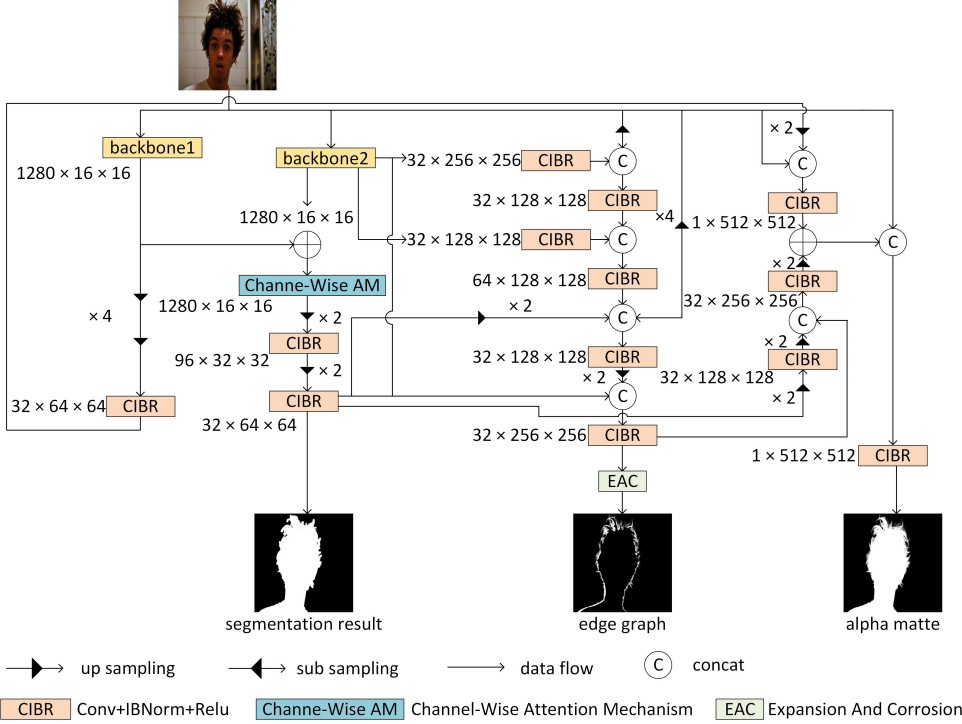

**Figure 2.** A diagram of a static teacher network as the complex structure.

### 3.1.2. Channel-Wise Attention Mechanism

To capture the importance of each channel in the feature map and enhance the subsequent processing of higher-importance channels, we designed this attention mechanism. In this attention mechanism, we calculate the importance value of each channel in the feature and establish the relationship with the feature. The specific implementation steps are shown in Algorithm 1 and a simple illustration is shown in Figure 3. First of all, we assume that the input is a single image, and the batch size is 1 at this time. We ignore the batch dimension, so the feature shape can be represented as [C,16,16]. Secondly, we evaluate each of the channels in the feature through average pooling, and by doing that the feature will translate into a tensor of only C length. Thirdly, we reduce the dimension of the features extracted in the previous step to one-quarter of the original by convolution and then restore it to the original length C after activation of the ReLu [36] function. Fourthly, we use the Sigmoid [37] function to transform a value of length C to between 0 and 1 and extend it by copying it to the same size as the original feature. By performing such an operation, we obtain some importance degree values greater than zero and less than one

for each channel. Finally, we multiply the features of the original input with the results of the previous calculation. This multiplicative approach amplifies the values in the channels with higher importance values, enhancing the visual color of these areas and the attention to subsequent processing. In summary, in the first four steps above we are trying to obtain an importance value between 0 and 1 for each channel in the feature, and then give a larger eigenvalue to the channel with a higher value by multiplying the original feature. Therefore, we can utilize the channel-wise attention mechanism to calculate the importance of each channel feature and pay more attention to channels with higher importance.

---

**Algorithm 1:** The algorithm

---

1 of the Channel-Wise Attention Mechanism
  **Input:** The features F.
  **Output:** Features with varying degrees of attention.
2 The channel-wise Attention Mechanism first establishes the calculation of the importance values of different channels and then correlates them with the features of the original input. Suppose that the shape of our input feature F is [B, C, H, W]. View stands for morphing the features. Conv is used to represent the convolution operation. Expand indicates that a column of values is copied to achieve matrix expansion.
3 **for** *i=1 to W step 1* **do**
4     **for** *j=1 to H step 1* **do**
5         | P_sum += F(i,j)
6     **end**
7 **end**
8 P_value $\leftarrow \frac{1}{H \times W}$ P_sum
9 P_view_1 $\leftarrow$ Conv(View(P_value)) // Channel $\longrightarrow$ Channel/4
10 **for** *j=1 to C/4 step 1* **do**
11     value $\leftarrow$ P_view_1$_j$
12     **if** *value >0* **then**
13         | value $\leftarrow$ value
14     **end**
15     **else**
16         | value $\leftarrow \alpha(e^x$-1)
17     **end**
18     P_view_1$_j \leftarrow$ value
19 **end**
20 P_view_2$_j \leftarrow Conv(P\_view\_1_j)$ // Channel/4 $\longrightarrow$ Channel
21 **for** *j=1 to C step 1* **do**
22     value $\leftarrow$ P_view_2$_j$
23     value $\leftarrow \frac{1}{1+e^{-x}}$
24     P_view_2$_j \leftarrow$ value
25 **end**
26 P_view_3$_j \leftarrow$ View(P_view_2$_j$) // [C] $\longrightarrow$ [C,1,1]
27 output $\leftarrow$ F $\times$ Expand(P_view_3$_j$) // [C] $\longrightarrow$ [C,16,16]

---

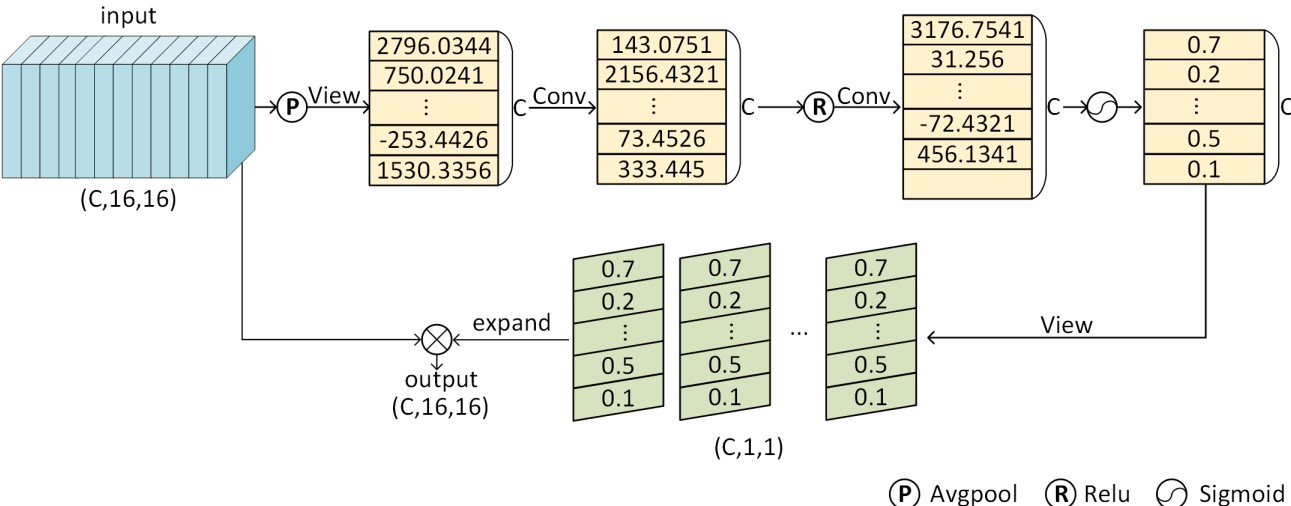

**Figure 3.** A diagram of the channel-wise attention mechanism.

### 3.1.3. CIBR

It is generally accepted that BatchNorm [38] is suitable for cognitive tasks (classification and segmentation, etc.), while instanceNorm [39] is suitable for generative tasks (style shift, Deblur, GAN). When you change the model's BatchNorm to instanceNorm during a cognitive task, a significant performance drop is shown; however, this is not because instanceNorm is not friendly to cognitive tasks. Through our experiments, a different implementation can avoid performance degradation. In fact, InstanceNorm and Batchnorm each have unique advantages. InstanceNorm brings immutability to the statistical variance across different samples, which is reflected in the characteristic mean variance statistics. BatchNorm, as a global normalization method, preserves the differences among samples in the datasets. InstanceNorm brings invariance that Batchnorm cannot replace. For example, the variance generated by the input image on appearance factors (such as brightness, color, style, etc.); however, InstanceNorm also destroys significant differences between different categories in the BatchNorm. However, in portrait matting, the effect of the network depends on the differentiation of different types of objects in the feature maps. This variation is largely reflected in mean and variance, and we do not want instanceNorm to destroy this divergence. Therefore, to better benefit from the advantages of BatchNorm and InstanceNorm, IBNorm is proposed by us and its effect was verified in experiments.

IBnorm refers to the following process: assume that features of the format are represented as [batchsize, channel, height, width]. The features are first processed by BatchNorm and then passed to InstanceNorm for processing. Finally, the characteristics are stacked on the first dimension. IBnorm is embedded in our proposed CIBR module to replace the traditional CBR design. CBR means embedding Batchnorm between convolution and ReLU in existing code. In the CIBR module that we designed, the convolutional layer is followed by IBnorm processing. Finally, the activation function ReLU function is adopted to increase the nonlinearity of the network.

### 3.1.4. The Process of Data Transmission

In Figure 2, we introduce the formation process of segmentation results, edge graphs, and alpha mattes. These three parts play an important role in the network, the main functions are: (1) As the standard for calculating loss measurement between static teacher network and static student network when training labeled dataset. (2) As a reference to guide the static student network when there are no labeled datasets in training. (3) As a training standard for the adaptive student network. In the above three outputs, the edge graphs need to be further obtained by expansion and corrosion (Abbreviated as EAC in Figure 2) operations on the obtained feature graph: the intersection area of foreground

expansion and corrosion is obtained as the edge graph at this time. These edge graphs are obtained for later training to compare edge effects.

In the following part, we introduce how to realize loss calculation in the training process through segmentation results, edge graphs, and alpha mattes. It is assumed that the segmentation results, edge graphs, and alpha mattes of the loss to be calculated are expressed as $s_i$, $e_i$, and $a_i$, respectively. Here we give some formulaic representations of the calculation.

$$segloss = \frac{1}{2}||s_i - Gau(\alpha_m)||_2, \tag{1}$$

$$eloss = ||e_i - EAC(\alpha_m)||_1, \tag{2}$$

$$aloss = ||a_i - \alpha_m||_1, \tag{3}$$

$$Loss = \lambda_s segloss + \lambda_e eloss + \lambda_a aloss, \tag{4}$$

where $\alpha$ stands for alpha mattes in the datasets and Gau represents the process of downsampling and Gaussian blur. "$|| \ ||_1$", "$|| \ ||_2$" represent the L1 loss and L2 loss calculation [40], respectively. EAC is used to indicate expansion and corrosion operations on the features. In this paper, the size of the convolution kernel used for expansion and corrosion operations is 3. Equation (2) is designed to calculate the differences between the segmentation results generated by the S-TN network and the S-SN network. Equation (3) is adopted to measure the similarity of edge graphs obtained after expansion and corrosion operation changes in the S-TN network and S-SN. Equation (4) is responsible for calculating the gap between the outputs in the final S-SN network and the annotation results in the datasets. Equations (2) and (3) both attempt to assist training by calculating losses, so that the effect of S-SN with a relatively simple structure is close to that of the S-TN network with a relatively complex structure. The purpose of Gaussian blur [41] processing is to remove some hair details because at this step we only need to obtain the outline of the portrait as much as possible, and we do not need to pay attention to details. In addition, we conduct the calculation after downsampling in this step to reduce the time of calculation loss by reducing the number of parameters. The hyper-parameters $\lambda_s$, $\lambda_e$, and $\lambda_a$ in Equation (5) are originally set to 1, 10, and 1, respectively. Equations (2)–(5) are designed to calculate losses when using labeled data to train static teacher networks and static student networks. However, there is no alpha matte when it comes to training unlabeled data. When the static teacher network assists the adaptive student network in training unlabeled data, the alpha mattes from the data set in Equations (2)–(5) should be replaced with the alpha mattes generated in the static teacher network.

### 3.1.5. Pruning the Static Student Network

Our static student network (S-SN) and adaptive student network (A-SN) are transformed based on the static teacher network (S-TN). To obtain more abundant characteristics, two kinds of backbones are designed in our static teacher network. In the following process, the features of one backbone are utilized as residual edges to complement the features acquired by the other backbone; however, in the real application, the student network needs not to be as complex as the teacher network [20]. Therefore, we adjust the static teacher network to obtain a lightweight static student network. When adjusting, we simply remove backbone1 and its associated residual edges to achieve refactoring. Through experiments, it is proved that this simple student network can also obtain good portrait matting effects under the guidance of the teacher network.

### 3.2. Adaptive Strategies

In the following sections, we introduce two training strategies for unlabeled data. By adopting these two adaptive strategies, ideal models can be constructed even when training data are insufficient. It is worth noting that these two strategies are also significant innovations compared with the previous teacher–student models. These two strategies enable the network to not only rely on the output of the last layer of the static teacher network for optimization but also establish a reference to the characteristics of the middle layer when faced with unlabeled data. In addition, the standard referenced in the training

of the adaptive student network is not only from the static teacher network but also from their network.

### 3.2.1. Auxiliary Adaption

Auxiliary adaption can be adopted when static teacher networks guide adaptive student networks without labels [42–44]. Its core idea can be described as using a small number of labeled datasets to train the static teacher network and static student network. Then, the trained model of the static student network is copied to the adaptive student network. In the face of unlabeled datasets, the trained static teacher network with a complex structure is first used to obtain segmentation results, edge graphs, and alpha mattes. It is assumed that alpha mattes generated by the static teacher network are expressed as $a_m$. Then, this batch of data is transmitted to the adaptive student network. It is assumed that the segmentation results, edge graphs, and alpha mattes obtained by the student network are represented by $s_i$, $e_i$, and $a_i$, respectively. Finally, Equations (2)–(5) are adopted for the learning and training of the adaptive student network. This adaptive strategy successfully establishes a connection between the static teacher network and the adaptive student network and enables the relatively simple adaptive student network to achieve a huge improvement in the training effect on the unmarked pictures.

### 3.2.2. Self-Adjusting Adaption

Self-adjusting adaption is designed to establish supervised learning between characteristics of different layers in the adaptive student network. With the help of self-adjusting adaption, the student network can reduce absolute dependence on the static teacher network [20,45,46]. In the adaptive student network, we retain the same three output results as the teacher network, namely, segmentation results, edge graphs, and alpha mattes. Formally, we denote our adaptive student network as A-SN and represent the three types of output as $\tilde{s}$, $\tilde{e}$, and $\tilde{a}$, respectively. Alpha mattes generated in the deep part of the network are more meticulous than segmentation results and edge graphs obtained in the front end of the network. Thus, alpha mattes serve as reference points for guided learning in this adaptive strategy. When implementing self-adjusting adaption, network optimization follows the following equation:

$$Loss\_saa = ||Gau(\tilde{\alpha}) - \tilde{s}||_2 + ||EAC(\tilde{\alpha}) - \tilde{e}||_1. \tag{5}$$

In this way, we enable the network to obtain the effect of the deep network as much as possible when there are fewer layers in the early stage, to further optimize the details in the deep network.

## 4. Experimental Results

In this section, we describe the experimental implementation in detail and report our portrait matting results on different benchmarks compared with other algorithms. Finally, the effects of adaptive strategies and self-adjusting adaption are investigated through ablation studies.

### 4.1. Self-Made Dataset

After reviewing the public datasets, we find that the number of datasets in the portrait matting domain is smaller than in other computer vision domains. In addition, most of the people in the existing datasets are non-Asian. To enrich the diversity of datasets in this field, a total of 130 sets of data are finely annotated by us (as shown in Figure 4). Compared with the existing data set, we have the following three differences: (1) Our dataset complements the diversity of race. It provides convenience for the following scholars to conduct research. (2) We add multi-person images to our data set to avoid the phenomenon of only one person in the previous dataset. (3) Some images in poor lighting conditions are also taken and annotated. We name our dataset a multi-category portrait matting dataset (MPMD) and will make it public for scholars to use.

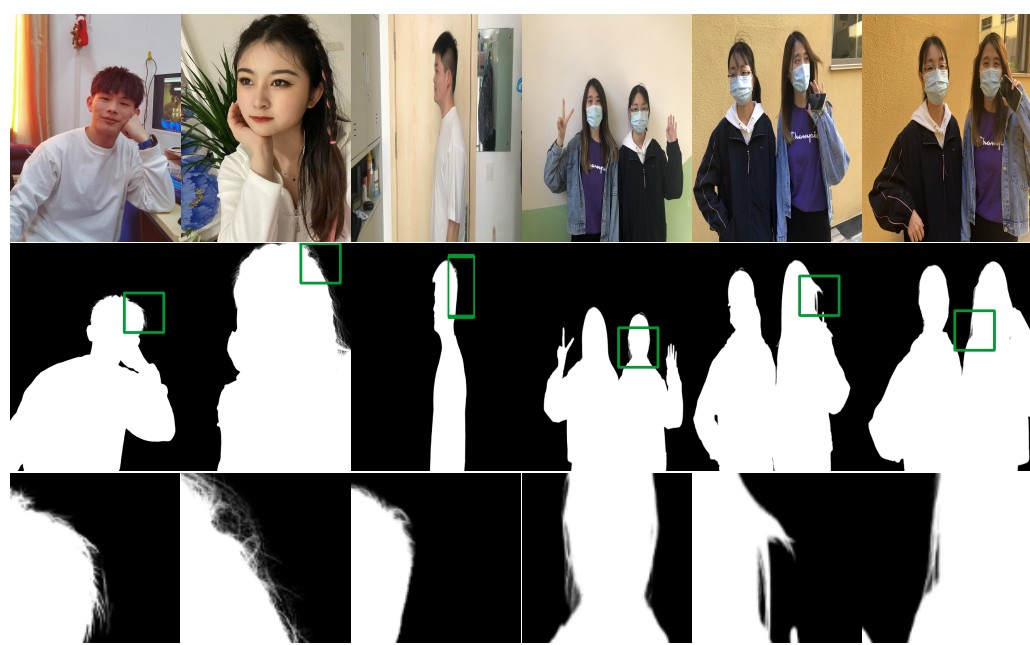

**Figure 4.** A presentation of some images of our multi-category portrait matting dataset (MPMD).

**Table 1.** Comparisons of different methods on ImageNet.

| Method | ImageNet [47] | |
| --- | --- | --- |
| | FLOPs | Accuracy (%) |
| FBNet [48] | 375M | 74.9 |
| ProxylessNAS [49] | 465M | 75.1 |
| ChamNet [50] | 553M | 75.4 |
| ResNet [51] | 600M | 75.5 |
| MobileNetV3 [31] | 356M | 76.6 |
| EfficientNet [52] | 390M | 77.3 |
| AtomNas [53] | 363M | 77.6 |
| FBNetV2 [30] | 423M | 78.1 |

### 4.2. Training Details

We first adopt SPDDataset [54] and the training sets of AutomaticPortraitMattingDataset [55] to train the static teacher network and static student network, respectively, with 800 epochs. The initial learning rate is set at 0.03 and shrunk by 0.45 times after every 100 epochs. Moreover, adam [56] is chosen as our optimized way. After the training, we directly copy the static student network parameters to the adaptive student network model. Then, the adaptive student network is fed a batch of unlabeled data and adjusts parameters according to auxiliary adaption and self-adjusting adaption. Finally, several images from different datasets are tested to verify the effect. The names and number of datasets we utilize for testing and training are summarized and presented in Table 2.

**Table 2.** Dataset description for portrait matting.

| Datasets | PPM-100 [46] | SPDDataset [54] | Adobe Portrait Matting Dataset [57] | Automatic Portrait Matting Dataset [55] | P3M-10k [58] |
| --- | --- | --- | --- | --- | --- |
| Train Samples | 0 | 3210 | 0 | 1700 | 0 |
| Test Samples | 100 | 3210 | 636 | 300 | 1000 |
| Testset Rename | - | SPDDataset | AdobeDataset | AutomaticDataset | P3M |

### 4.3. Comparisons Experiments

In this section, the details and effects of some comparative experiments are introduced. The content includes: (1) the effect compared with the recent portrait matting architectures. (2) The effect comparison of some previous modules with a similar design to CIBR. (3) The effect comparison of S-TN and S-SN designed by us. (4) The parameter comparison of S-SN designed by us and the recent portrait matting architectures.

#### 4.3.1. Comparisons of State-of-the-Art Methods

To verify the effectiveness of our network architecture, multiple comparative experiments were performed and the results are recorded in Table 3. SAD, MSE, MAD, Grad, and Conn are the abbreviation of the 'sum of absolute differences', 'mean squared error', 'mean absolute difference', 'grad error', and 'connectivity error'; SAD-FG/SAD-BG is: the sum of absolute differences in the foreground/background. We conducted comparative tests on four different datasets, namely PPM-100 [46], SPDDataset [54], AdobeDataset [57], and AutomaticDataset [55]. Five methods recently applied in portrait matting were selected as the targets of our comparative experiments. It is worth noting that there are no open-source codes for the previous architectures that adopt the idea of knowledge distillation, so we have not compared it to these types of approaches. To ensure the fairness of model comparison, we first trained 800 epochs on the training sets of SPDDataset [54] and AutomaticDataset [55] for the architectures of the comparative experiments. The initial learning rate was set at 0.06, and after 80 epochs each time, the learning rate decreased to half of the original. At the same time, the Adam optimizer [56] was applied as the method of training optimizer. Seven types of different calculations were chosen to measure the effectiveness of the comparative tests. The seven calculation methods used to measure the experimental effect include mean squared error (MSE), meaning absolute difference (MAD), solute differences (SAD), gradient (Grad), Connectivity (Conn), SAD-FG, and SAD-BG. The calculation methods are consistent with that mentioned in the previous paper [59].

The results of 24 groups of comparative experiments can be viewed in Table 3. Experiments show that our network architecture can achieve good results on multiple test sets. In particular, in our comparative experiments with the AdobeDataset [57], our architecture achieves significant improvements over the recent approach named SPKD [60] in several metrics. The generalization ability of ASSN is fully validated by testing on different untrained datasets. In addition, some visual comparisons are shown in Figure 5.

**Table 3.** The results of our approach compared with recent architectures on four different datasets.

| Dataset | PPM-100 [46] | | | | | | SPDDataset [54] | | | | | |
|---|---|---|---|---|---|---|---|---|---|---|---|---|
| Method | GFM [59] | P3M [58] | MGM [61] | ViTAE [62] | SPKD [60] | ASSN | GFM [59] | P3M [58] | MGM [61] | ViTAE [62] | SPKD [60] | ASSN |
| year | 2022 | 2021 | 2021 | 2022 | 2020 | 2022 | 2022 | 2021 | 2021 | 2022 | 2020 | 2022 |
| SAD | 52.00 | 65.39 | 34.06 | 32.91 | 35.29 | 20.86 | 49.69 | 23.57 | 22.02 | 19.27 | 21.31 | 18.21 |
| MSE | 0.1936 | 0.1973 | 0.0502 | 0.0761 | 0.0699 | 0.0750 | 0.1868 | 0.0681 | 0.0501 | 0.3107 | 0.0538 | 0.0673 |
| MAD | 0.1983 | 0.2491 | 0.1293 | 0.1256 | 0.1967 | 0.0790 | 0.1895 | 0.0952 | 0.0771 | 0.3244 | 0.0865 | 0.0695 |
| Grad | 19.07 | 18.98 | 14.48 | 16.46 | 14.04 | 17.09 | 27.21 | 23.18 | 12.72 | 23.42 | 12.91 | 19.89 |
| Conn | 50.30 | 65.81 | 32.87 | 32.99 | 36.94 | 21.23 | 49.56 | 21.72 | 19.42 | 17.92 | 20.05 | 10.59 |
| SAD-FG | 36.37 | 20.44 | 12.08 | 13.91 | 12.33 | 10.34 | 32.11 | 5.225 | 6.174 | 5.310 | 4.061 | 10.50 |
| SAD-BG | 7.727 | 10.36 | 10.59 | 8.204 | 12.09 | 6.223 | 7.577 | 14.05 | 15.09 | 10.94 | 14.59 | 2.926 |

| Dataset | AdobeDataset [57] | | | | | | AutomaticDataset [55] | | | | | |
|---|---|---|---|---|---|---|---|---|---|---|---|---|
| Method | GFM [59] | P3M [58] | MGM [61] | ViTAE [62] | SPKD [60] | ASSN | GFM [59] | P3M [58] | MGM [61] | ViTAE [62] | SPKD [60] | ASSN |
| year | 2022 | 2021 | 2021 | 2022 | 2020 | 2022 | 2022 | 2021 | 2021 | 2022 | 2020 | 2022 |
| SAD | 77.09 | 28.54 | 25.37 | 25.62 | 26.08 | 21.79 | 60.77 | 47.58 | 45.03 | 47.27 | 45.53 | 31.68 |
| MSE | 0.2839 | 0.2131 | 0.0676 | 0.1072 | 0.0706 | 0.0734 | 0.2214 | 0.2151 | 0.1765 | 0.1072 | 0.1795 | 0.1118 |
| MAD | 0.2940 | 0.3325 | 0.2105 | 0.2613 | 0.2208 | 0.0831 | 0.2318 | 0.2195 | 0.1912 | 0.1280 | 0.2063 | 0.1208 |
| Grad | 24.13 | 20.22 | 16.60 | 17.58 | 16.97 | 17.38 | 17.89 | 15.33 | 12.62 | 13.59 | 13.06 | 9.696 |
| Conn | 70.17 | 28.18 | 24.85 | 25.21 | 25.02 | 20.77 | 59.62 | 47.43 | 44.19 | 24.44 | 44.67 | 39.37 |
| SAD-FG | 21.66 | 12.10 | 5.019 | 2.311 | 10.64 | 2.077 | 52.14 | 28.69 | 22.62 | 14.80 | 23.19 | 25.00 |
| SAD-BG | 15.47 | 9.242 | 3.628 | 2.870 | 3.766 | 2.848 | 2.569 | 2.815 | 4.180 | 6.392 | 3.547 | 1.954 |

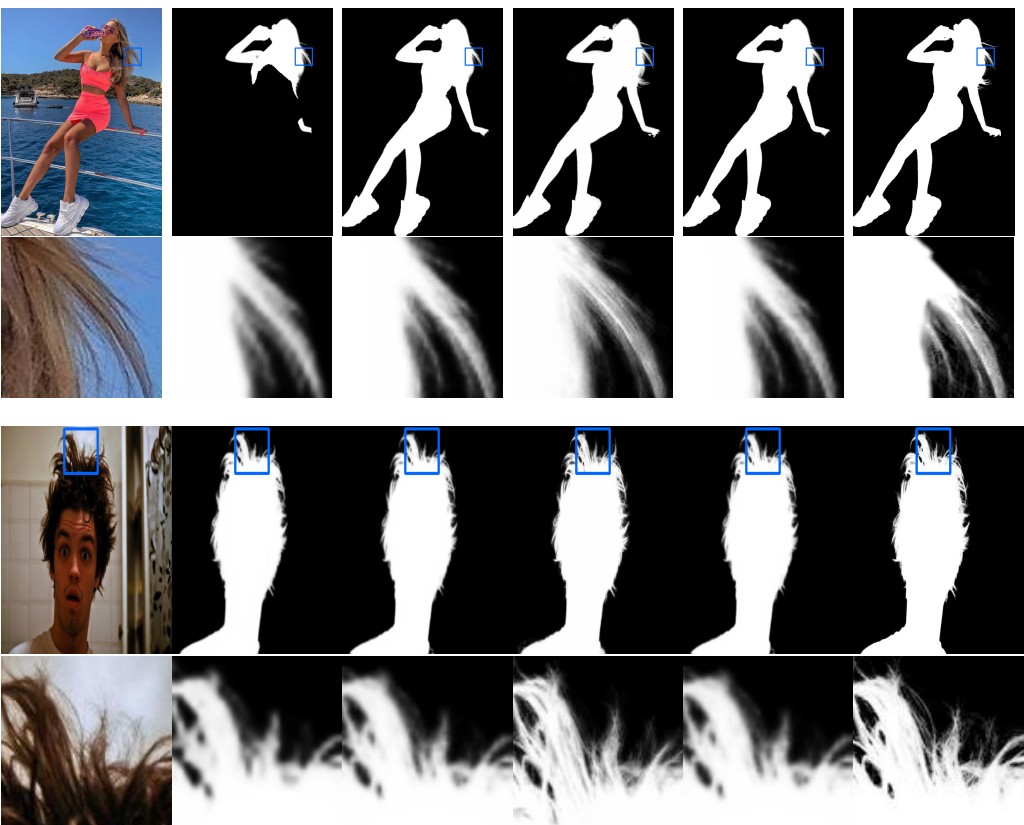

**Figure 5.** Visual comparisons for portrait matting. From left to right, the above figures show the original images, the results of GFM [59] architecture, the results of P3M [58] architecture, the results of MGM [61], and the results of ViTAE [62] and the results of our architecture.

4.3.2. CIBR vs. CBR, CIR, and CBIR

The difference between CIBR, CBR, CIR, and CBIR is that they are embedded in different normalized ways between the convolutional layer and the ReLU activation function. What CIBR chose is to use InstanceNorm first and then BatchNorm. CBR implies the BatchNorm as the method of normalization. CIR stands for the adoption of InstanceNorm. CBIR uses BatchNorm first and then InstanceNorm. More details about each of these modules are shown in Section 3.1.3. Some relevant experimental results are presented in Table 4. To compare the different results brought by these different designs embedded in our network, we conducted comparative tests on these modules. During the experiment, we replaced the "CIBR" in our architecture with "CBR", "CIR", and "CBIR", respectively, and carried out training. The experiments all underwent the same training strategy before the test. All tests were performed on the PPM-100 dataset [46]. The datasets, training epochs, learning rates, and optimizers used for training are the same as described in Section 4.2. The experimental data in Table 4 further confirm that simply embedding Instancenorm is not a wise choice in the architecture. It was 5.44, 0.0266, 0.0247, 1.82, 0.41, 1.85, and 6.346 higher than SAD, MSE, MAD, Grad, Conn, SAD-FG, and SAD-BG when using Batchnorm directly. In addition, when Instancenorm and Batchnorm are embedded at the same time, CIBR design is superior to CBIR in measuring indicators. Through these sets of experiments, we effectively prove that using "CIBR" modules in our architecture is more conducive to portrait matting than using the other three designs.

**Table 4.** Comparison Results Related to CIBR.

| Module | CIBR | CBR | CIR | CBIR |
|--------|------|-----|-----|------|
| SAD | 20.86 | 29.93 | 35.37 | 23.18 |
| MSE | 0.0750 | 0.0757 | 0.1023 | 0.0805 |
| MAD | 0.0790 | 0.1072 | 0.1319 | 0.0937 |
| Grad | 17.09 | 21.33 | 23.15 | 20.92 |
| Conn | 21.23 | 22.50 | 32.91 | 21.42 |
| SAD-FG | 10.34 | 12.72 | 14.57 | 13.81 |
| SAD-BG | 6.223 | 8.184 | 14.53 | 7.106 |

### 4.3.3. The Comparisons of Static Teacher Network and Static Student Network

In this paper, we try to guide the learning and training of static student networks (S-SN) by static teacher networks (S-TN) with complex structural design. The purpose of this behavior is to enable the relatively simple structure of the static student network to have similar effects to the complex structure of the teacher network. In this part, we compare the effects of the static teacher network and the static student network on the PPM-100 dataset [46]. It is important to note that the adaptive student network is acquired by copying the static student network to face the unlabeled datasets with the help of adaptive strategies. So in this subsection, the effect of the adaptive student network is the same as that of the static student network. We only need to compare the effect of S-TN and S-SN, which is enough to confirm whether the relatively simple S-SN achieves the same effect as S-TN. By testing S-TN, we obtain the results of 20.91, 0.0751, 0.0792, 17.13, 21.25, 10.39, and 6.226 on SAD, MSE, MAD, Grad, Conn, SAD-FG, and SAD-BG, respectively. Only 0.05, 0.0001, 0.0002, 0.04, 0.02, 0.05, and 0.003 are lower than S-SN on the seven measures, respectively; therefore, we succeed in making S-SN with a relatively simple structure under the guidance of S-TN to achieve similar results.

### 4.3.4. Parameters

To verify that the structure we designed is relatively simple, the number of parameters for multiple architectures were tested and recorded. Through the test, we learn that the number of parameters of GFM, P3M, MGM, and ViTAE is 55.29M, 39.48M, 29.7M, and 27.46M, respectively. Our architecture has only 6.49M parameters. The number is much smaller than that of some recent architectures.

### 4.4. Ablation Studies

In this section, we introduce some ablation experiments performed to verify the effectiveness of the designed modules and adaptive strategies.

### 4.4.1. Adaptive Strategies

In Section 3.2, we introduce two types of innovative adaptive strategies. To verify the effectiveness of the proposed adaptive strategies, we combined the adaptive strategies with several different current architectures and tested their effects. Test results before embedding the adaptation strategies are recorded in columns 2 through 6 of Table 5. The testing effects after the adaptive strategies are integrated based on different architectures, and are shown in the last five columns of Table 5. First, we trained 400 epochs on the training set of SPDDataset [54]. The initial learning rate was set as 0.06, and after 80 epochs each time, the learning rate decreased to half of the original. At the same time, we used the Adam optimizer [56] to optimize models during training. Then, the tests were performed on the P3M test set [58], and the results of the tests are recorded in the first four columns of Table 5. Finally, four groups were trained using the two adaptive strategies we designed and the test results are recorded in the last four columns of the table. According to the comparison between the data in the first four columns of the table and the data in the next four columns, it is obvious that the adaptive strategy designed by us can be significantly improved in the test set. After adding the adaptive strategy designed by us, the recently proposed MGM [61] architecture decreases by 29.56, 0.1131, 0.1151, 24.6, 27.03, 9.7562, 11.46 in SAD, MSE, MAD, Grad, Conn, SAD-FG, and SAD-BG, respectively. At the same time, although

MSE and MAD have obvious advantages over other methods when ViTAE architecture [62] does not use our adaptive strategies, they still improve significantly after training using the two adaptive strategies we designed. Among them, MSE and MAD further decreased by 0.0850 and 0.0827, respectively. These eight sets of experiments effectively demonstrate the two adaptive strategies designed by us can improve the generalization ability of the existing architectures even after training on small batch datasets.

**Table 5.** Comparison results with adaptive strategies.

| Method | Before (the Original Algorithms) | | | | | After (Algorithms with the Adaptive Strategies) | | | | |
|---|---|---|---|---|---|---|---|---|---|---|
| | GFM [59] | P3M [58] | MGM [61] | ViTAE [62] | Ours w/o Strategies | GFM [59] | P3M [58] | MGM [61] | ViTAE [62] | Ours |
| SAD | 66.50 | 30.74 | 35.21 | 23.75 | 17.59 | 43.72 | 4.520 | 5.641 | 2.368 | 2.074 |
| MSE | 0.2500 | 0.1071 | 0.1246 | 0.0898 | 0.0648 | 0.1620 | 0.0125 | 0.0115 | 0.0048 | 0.0031 |
| MAD | 0.2536 | 0.1172 | 0.1366 | 0.0917 | 0.0706 | 0.1668 | 0.0172 | 0.0215 | 0.0090 | 0.0075 |
| Grad | 20.22 | 33.31 | 28.42 | 29.91 | 20.59 | 22.67 | 6.876 | 3.820 | 4.838 | 4.575 |
| Conn | 64.86 | 36.89 | 32.06 | 22.41 | 10.93 | 42.96 | 4.276 | 5.027 | 2.249 | 2.084 |
| SAD-FG | 54.79 | 13.04 | 10.63 | 8.635 | 9.42 | 33.07 | 1.083 | 0.8738 | 0.3509 | 0.3018 |
| SAD-BG | 2.436 | 6.303 | 14.81 | 10.62 | 2.834 | 2.503 | 0.9829 | 3.341 | 0.1132 | 0.0948 |

### 4.4.2. Some Components in the Teacher Network

In the static teacher network, a total of four components are embedded. They are backbone1, backbone2, channel-wise attention mechanism (Channel-Wise AM), and CIBR modules. In this section, we combine these four components and test them on the P3M test datasets [58]. Twelve sets of ablation experiments are presented in Table 6. In addition, the twelve ablation groups underwent the same training strategy on the same dataset. The training period, dataset, learning rate, and selection of optimizer are the same as the configuration in Section 4.2.

**Table 6.** The ablation experiment of blocks in the static teacher network.

| Num | backbone1 | backbone2 | Channel-Wise AM | CIBR | SAD | MSE | MAD | Grad | Conn | SAD-FG | SAD-BG |
|---|---|---|---|---|---|---|---|---|---|---|---|
| 1 | ✓ | ✗ | ✗ | ✗ | 54.25 | 0.1256 | 0.1726 | 18.72 | 53.21 | 42.06 | 9.825 |
| 2 | ✗ | ✓ | ✗ | ✗ | 60.94 | 0.2355 | 0.2286 | 21.48 | 54.28 | 42.85 | 8.572 |
| 3 | ✓ | ✓ | ✗ | ✗ | 42.57 | 0.1547 | 0.1703 | 23.83 | 41.64 | 20.58 | 7.852 |
| 4 | ✓ | ✗ | ✓ | ✗ | 50.17 | 0.1102 | 0.1653 | 14.95 | 49.83 | 38.01 | 9.194 |
| 5 | ✓ | ✗ | ✗ | ✓ | 46.82 | 0.1053 | 0.1493 | 11.15 | 43.03 | 33.92 | 8.083 |
| 6 | ✗ | ✓ | ✓ | ✗ | 56.28 | 0.1738 | 0.1841 | 17.74 | 50.92 | 39.61 | 7.440 |
| 7 | ✗ | ✓ | ✗ | ✓ | 48.67 | 0.1383 | 0.1618 | 13.47 | 46.92 | 30.04 | 7.052 |
| 8 | ✗ | ✓ | ✓ | ✓ | 26.30 | 0.0961 | 0.1075 | 10.31 | 23.72 | 14.10 | 7.228 |
| 9 | ✓ | ✗ | ✓ | ✓ | 13.94 | 0.0619 | 0.0690 | 16.36 | 9.186 | 2.330 | 4.124 |
| 10 | ✓ | ✓ | ✗ | ✓ | 8.813 | 0.0135 | 0.0171 | 5.446 | 8.350 | 0.9182 | 1.843 |
| 11 | ✓ | ✓ | ✓ | ✗ | 10.72 | 0.0979 | 0.0893 | 20.49 | 10.18 | 3.483 | 6.421 |
| 12 | ✓ | ✓ | ✓ | ✓ | 2.074 | 0.0031 | 0.0075 | 4.575 | 2.084 | 0.3018 | 0.0948 |

It is worth mentioning that channel-wise AM and CIBR require feature input for further feature extraction; therefore, there is no need to set up experiments utilizing channel-wise AM and CIBR without any backbone. However, through the comparison of several experiments, it is proved that adding channel-wise AM or CIBR can indeed improve the effect of portrait matting. After adding the channel-wise AM structure based on backbone1, SAD, MSE, MAD, Grad, Conn, SAD-FG, and SAD-BG decrease by 4.08, 0.0154, 0.0073, 3.77, 3.38, 4.05 and 0.631, respectively. When backbone1 is integrated with CIBR, the seven measures fall even more sharply. They decreased by 7.43, 0.0203, 0.0233, 7.57, 10.18, 8.14, and 1.742, respectively. From the comparison of group 2 and group 6 in Table 6, it can be found that the combination of backbone2 with channel-wise AM or CIBR can also significantly improve the effect. As can be seen from the comparison results of group 2 and group 7, the use of CIBR can also further improve the effect of backbone2.

As far as the use of backbone is concerned, the effect of the combination of the two is better than that of the single embedding. When adopting the combination with

two backbones, such as the second group and the third group of experimental comparison, the SAD, MSE, and MAD index decreased by 18.37, 0.0808, and 0.0583 respectively. In general, channel-wise AM or CIBR embedding can enhance the effect of the original architecture. In addition, the design of two backbone companies is more effective than using only one backbone company.

## 5. Conclusions

In this paper, we propose two kinds of novel adaptive strategies and a semi-supervised network. The two adaptive strategies are auxiliary adaption and self-adjusting adaption. These two adaptive strategies assist the semi-supervised network to have the ability to further improve its effectiveness in the face of unlabeled datasets. In our proposed architecture, there are three parts: static teacher network (S-TN) with a complex network structure, static student network (S-SN) acquired after pruning the teacher network, and adaptive student network (A-SN) applied to unlabeled datasets. Auxiliary adaption provides training and optimization reference for the A-SN through segmentation results, edge graphs, and pseudo-labels obtained from S-TN. This strategy eliminates time-consuming manual annotation in the face of unlabeled training datasets and improves the model effect on unlabeled datasets to a certain extent. The self-adjusting adaption enables the A-SN to measure and optimize the results generated in front of the network according to the results generated in the back layer of its network. In addition, we specifically integrated the above ideas with a network called ASSN and conducted 24 groups of comparison experiments on four datasets. To verify the effectiveness of the two adaptive strategies proposed by us, four different ablation experiments were performed by combining the existing four different architectures with our adaptive strategies. We also conducted some ablation experiments based on the modules in the designed network. Both theoretically and experimentally, our proposed architecture and two adaptive strategies are well proved. At the same time, considering that the existing detailed datasets in the field of portrait matting are not sufficient, dataset named MPMD is available for scholars to use. Compared to the previous datasets, our dataset not only adds images from multiple people but also complements the images from bad lighting. Because our architecture is capable of obtaining relatively detailed alpha mattes even on unlabeled datasets, it can be used to create some supplementary alpha mattes to enrich the reference of unlabeled datasets during training. In addition, our static teacher network can be used to guide lightweight networks for knowledge distillation-based learning.

**Author Contributions:** Conceptualization, G.W. and C.C.; methodology, X.Z.; software, X.Z.; validation, X.Z., and H.D.; formal analysis, H.D.; investigation, X.Z.; resources, G.W.; data curation, X.Z.; writing—original draft preparation, M.S.; writing—review and editing, X.Z.; visualization, M.S.; supervision, G.W.; project administration, G.W.; funding acquisition, C.C. All authors have read and agreed to the published version of the manuscript.

**Funding:** This research was funded by the Youth Innovation and Technology Support Plan of Colleges and Universities in Shandong Province OF FUNDER grant number 2021KJ062.

**Institutional Review Board Statement:** Not applicable.

**Informed Consent Statement:** Not applicable.

**Data Availability Statement:** The dataset(MPMD) can be found at https://github.com/XinyueZhangqdu/ASSN.

**Acknowledgments:** Youth Innovation and Technology Support Plan of Colleges and Universities in Shandong Province (2021KJ062).

**Conflicts of Interest:** The authors declare no conflict of interest. The funders had no role in the design of the study; in the collection, analyses, or interpretation of data; in the writing of the manuscript; or in the decision to publish the results.

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
