# Peer review of "Semi-Supervised Portrait Matting via the Collaboration of Teacher–Student Network and Adaptive Strategies"

_electronics, doi:10.3390/electronics11244080_

Round 1

Reviewer 1 Report

I reviewed the manuscript tittle Semi-supervised Portrait Matting via the Novel Teacher-Student Adaptation

The paper is interesting and discuss an import topic for many parties and beneficiaries

Abstract needs some more work and restructure with smooth flow of information to reflect the paper content clearly and to be shorten as it Is a bit long

I suggest that the Introduction part should not have formulas and equations, the introduction needs some reorganization, see for example

I = αF + (1 − α)B, (1)

And the static network diagrams, please move these to the heard of the paper or in the methods part, but for sure not the introduction. For example, you can place that after related work

Figure 2, and 3 has too long caption and needs some more clarification

Figure 2. This network with a complex structure is utilized to assist the student network to obtain optimization reference criteria including segmentation results, edge graphs, and alpha mattes when there is no label data

Figure 3. The diagram of the channel-wise attention mechanism. Calculate some importance values between 0 and 1 for each channel, and pay higher attention to channels with higher importance values.

The same is true for figure 4

Figure 4. A presentation of some images of our multi-category portrait matting dataset (MPMD). Our dataset includes single-person data as well as multi-person data that does not exist in the existing data set. In particular, we photographed and annotated the data under low light conditions. This dataset will be made available to scholars in the field of portrait matting

Section Backbone.( In our training network, we utilize the layout of two backbones.)

  What are they, please name them before moving forward

2.2.1. Auxiliary Adaption

This part needs some support and evidence from literature

And the same is true for this part

2.2.2. Self-adjusting Adaption 322

Table 2 has a tittle of one paragraph ??

Please shorten and provide the information after the table

Table 2. Comparision Results. SAD, MSE, MAD, Grad, and Conn are the abbreviation of the ‘sum of absolute differences’, ‘mean squared error’, ‘mean absolute difference’, ‘grad error’, and ‘connectivity error’; SAD-FG/SAD-BG is: the sum of absolute differences in the foreground/background

The combustion part is clear and promising, however, providing some implication for these conclusions will help reader and others to unitize your knowledge further in different places,

Authors might provide some implications and possible applications for their results and conclusions  

Reviewer 2 Report

This article presents very well-described research where the clearly defined originality and contribution are followed by a logical sequence of sections where the reader will find everything needed to understand the experiments and conclusions. On a positive note, the results are compared across a large number of metrics, the baseline method is clearly defined, and pseudocode and detailed workflow are available to understand the functionality.

Only minor weaknesses can be pointed out, namely:

- Please don't use too many abbreviations in headings (especially if the heading is only made of abbreviations, it's unprofessional)
- The article title and abstract are slightly misleading - it mentions a novel student-teacher adaptation, however, it sounds as if the student-teacher method is new here. The fact is that what is original is the overall framework of interconnected architectures derived from student-teacher systems, which doesn't entirely follow from the abstract, nor much from the described contributions of the paper.
- I would welcome perhaps a discussion on semi-supervised visual transformers - is it possible to use this new architecture here?
- Please describe in more detail in the descriptions of Tables 4 and 5. What are the results, what is in the table, etc...
- Section 3.4.1 - refers to section III part B, which cannot be found as this is typical IEEE formatting, not MDPI! Please check the referencing.
